# pH Sensitive Dextran Coated Fluorescent Nanodiamonds as a Biomarker for HeLa Cells Endocytic Pathway and Increased Cellular Uptake

**DOI:** 10.3390/nano11071837

**Published:** 2021-07-15

**Authors:** Linyan Nie, Yue Zhang, Lei Li, Patrick van Rijn, Romana Schirhagl

**Affiliations:** Department of Biomedical Engineering, University Medical Center Groningen, University of Groningen, A. Deusinglaan 1, 9713 AV Groningen, The Netherlands; niely0316@163.com (L.N.); yue.zhang@rug.nl (Y.Z.); l.li@umcg.nl (L.L.); p.van.rijn@umcg.nl (P.v.R.)

**Keywords:** nanodiamonds, cellular uptake, endocytic pathway

## Abstract

Fluorescent nanodiamonds are a useful for biosensing of intracellular signaling networks or environmental changes (such as temperature, pH or free radical generation). HeLa cells are interesting to study with these nanodiamonds since they are a model cell system that is widely used to study cancer-related diseases. However, they only internalize low numbers of nanodiamond particles very slowly via the endocytosis pathway. In this work, we show that pH-sensitive, dextran-coated fluorescent nanodiamonds can be used to visualise this pathway. Additionally, this coating improved diamond uptake in HeLa cells by 5.3 times (*** *p* < 0.0001) and decreased the required time for uptake to only 30 min. We demonstrated further that nanodiamonds enter HeLa cells via endolysosomes and are eventually expelled by cells.

## 1. Introduction

Fluorescent Nanodiamonds (FNDs) are carbon-based nanomaterials, which are widely used in the biomedical field. FNDs owe their fluorescence to specific defects in their lattice, the negatively charged nitrogen-vacancy colour-centres (the so-called NV centres). FNDs are perfectly photostable, do not photobleach, or blink. These features are perfectly suited for bioimaging applications [1]. The abundant chemical groups on the surface of FNDs including, for instance, alcohol, carbonyl, and carboxylic acid groups [2,3,4] in oxygen-terminated nanodiamonds allow surface modification and tailoring of properties.

Nanodiamonds have excellent biocompatibility in many different cell lines (in vitro) like HT-29 [5], macrophages, yeast cells [6], BHK-21 [7], MCF-7 [8]. Prabhakar et al. showed no toxicity after incubating MDA-MB-231 (human breast adenocarcinoma) cells with FNDs for 120 h. While other carbon nanomaterials have been reported to cause proinflammatory responses [9] this is not the case for nanodiamonds [10].

Additionally, no adverse effects have been observed in several in vivo experiments [11,12]. FNDs were, for instance, used for drug delivery in animal models like mice [13], fruit flies, worms, or pigs [14]. Chow et al. showed that the nanodiamond drug delivery system improved both drug retention in tumour cells and treatment safety and efficacy in murine cancer models for tumour treatment [15]. FNDs can be used for intracellular labelling since they allow long-term tracking [16]. Recently, it has also been demonstrated that nanodiamonds containing nitrogen-vacancy (NV) centres can be used to sense their magnetic surroundings with unprecedented sensitivity with nanoscale resolution [17,18]. When combined with pH-responsive polymers, their location can imply intracellular pH changes [19]. Charge conversion of NV centres can also be used as an indicator for pH but requires complex equipment for readout [20].

HeLa cells poorly internalize nanodiamond particles compared with other cell lines [21,22] (as, for instance, macrophages). Even after several hours of incubation there are not many diamond particles in HeLa cells. This observation poses a problem for researchers to use nanodiamonds as labels, for drug delivery or (to a lower extend) [15] for sensing.

For nanodiamonds and other nanoparticles, the initial interaction with cells is getting internalized. Surface charge or specific moieties in a coating influence cellular uptake or the intracellular fate of particles. Additionally, for FNDs several strategies to increase uptake have been demonstrated. Especially for larger cells, it is possible to inject FNDs [23]. There are also several techniques to treat cells chemically to induce nanodiamond uptake. Trypsin-EDTA treatment was used to separate colon cancer HT29 cells to increase interaction between diamonds and cells, leading to an increase in cellular uptake [24]. For yeast cells, which have a cell wall, the cell can be permeabilised chemically or the cell wall can be removed entirely [25]. However, these processes are quite invasive and thus not suited for all cell types or applications.

There are also several methods to alter the diamond surface to increase uptake including electropositive coatings [26], attaching molecules that are recognized by specific receptors [27], proteins [28] or nutrients.

With more diamond particles inside HeLa cells, researchers can further study intracellular diamond transportation. Knowing the location of diamonds can also be helpful in understanding drugs delivery by nanodiamonds as well as sensing [29,30,31].

Herein, we developed a new, fast, simple and straightforward strategy to increase nanodiamond uptake by HeLa cells and achieve pH sensitivity. With more diamond particles coated with a pH sensitive dye, we investigated the FNDs endocytosis pathway inside HeLa cells as well as the fate of diamond particles.

## 2. Materials and Methods

### 2.1. Materials

Fluorescent Nanodiamonds with a hydrodynamic diameter of 70 nm (FNDs70) containing >300 NV centres were purchased from Adamas Nanotechnologies, Brownleigh Drive Raleigh, NC, USA. They are produced from grinding larger high-pressure high temperature diamonds. Their surface is oxygen-terminated due to an acid cleaning treatment with oxidizing acids by the manufacturer. MTT assay kit, Live Cell Imaging Solution, pHrodo™ Green Dextran (10,000 MW for endocytosis) were purchased from ThermoFisher SCIENTIFIC, Bleiswijk, the Netherlands. LysoView™ 405 was ordered from Biotium to visualize lysosomes. Dextran with an average mol wt of 9000–11,000 extracted from the bacterial strain Leuconostoc mesenteroides was purchased from Sigma-Aldrich, Zwijndrecht, The Netherlands.

### 2.2. Cell Culture

HeLa cells were cultured in Dulbecco Modified Eagle Medium with 4.500 mg/mL glucose (DMEM-HG), supplemented with 10% Fetal Bovine Serum (FBS) and 1% penicillin/streptomycin at 37 °C and 5% CO_2_.

### 2.3. FND Characterization

A total of 50 µL dextran (1 mg/mL in H_2_O) was vortexed with 5 µL 70 nm FNDs (1 mg/mL in H_2_O) for 1 min. The mixture was further diluted in H_2_O to a final FND concentration of 5 µg/mL. A Malvern ZetaSizer Nanosystem (Malvern Instruments Ltd., Malvern, UK) was used to evaluate size distribution and surface charge of coated nanodiamonds.

### 2.4. SEM Imaging

Scanning Electron Microscopy (SEM) was performed using a Zeiss Supra55 ATLAS (University Medical Center Groningen, Groningen, The Netherlands) to obtain more details of pHrodo Green Dextran coating on the diamonds’ surface. FNDs mixed with pHrodo Green Dextran (FNDs final concentration 5 µL, pHrodo Green Dextran concentration 1 mg/mL) for 1 to 2 min, followed by 15 min incubation at room temperature to allow pHrodo Green Dextran adsorb FNDs. After incubation, pHrodo Green Dextran adsorb FNDs solution were transferred and dried on a silicon wafer, the samples were coated with a 5 nm gold layer, and imaged with a Philips XL30S SEM FEG instrument was used to image the samples at 5 kV. Bare FNDs were prepared as a control.

### 2.5. Cell Viability

HeLa cells were seeded (10,000 cells/well) in clear flat-bottom 96-well plates and incubated for 24 h until the confluency reached 70–90%. Then cells were incubated with 5 and 10 µg/mL FNDs, dextran (1 mg/mL) mixed with 5 and 10 µg/mL FNDs. After 24 h incubation, 20 µL MTT solution (0.75 μg/mL final concentration) was added into each well. The plates were further incubated at 37 °C for 3 h. Finally, the media were replaced with 200 µL dimethyl sulfoxide (DMSO). The absorbance at OD = 590 nm was determined using a synergy H1 microplate reader (BioTek). Untreated cells were used as a negative control. Cells treated with 5% DMSO were used as a positive control. Cell culture medium without cells was used as blank.

### 2.6. Cellular Uptake of FNDs70

Three different experimental groups were prepared: (1) cells were incubated with bare FNDs, (2) dextran mixed with FNDs (dextran concentration 1 mg/mL), or (3) pHrodo Green Dextran mixed with FNDs (pHrodo Green Dextran concentration 1 mg/mL) for 0.5 h or 4 h, respectively. The final concentration of FNDs in all groups was 5 µg/mL. Then, cells were fixed with 3.7% paraformaldehyde (PFA) for 10 min and stained with 4,6-diamidino-2-phenylindole (DAPI, for staining the nucleus), and Fluorescein phalloidin (FITC-phalloidin, for staining F-actin). Confocal images were acquired with a Zeiss 780 laser-scanning microscope (Zeiss, Jena, Germany). Nanodiamonds were detected at 561/659 nm, DAPI at 358/461 nm, and FITC at 495/510 nm. Confocal Z-stack images were analysed by a custom-made script using the 3D object counter [27] plugin of FIJI (http://fiji.sc/). The threshold was set to 41 (this number was determined earlier to be the value of an average particle for particles spread on a surface).

### 2.7. FNDs Subcellular Location

HeLa cells were seeded and cultured in 35 mm glass-bottom dishes overnight. Cells were first washed with phosphate-buffered saline (PBS, pH 7.4) 3 times. As a result of this treatment, cells are in a neutral extracellular environment (around pH 7.4). An amount of 2.5 µL of FNDs (1 mg/mL stock concentration) were mixed with 25 µL pHrodo Green Dextran in 500 µL DMEM high glucose (4500 mg/L) and added to cells. Then we incubated for 30 min, removed the medium, and washed cells with PBS (pH 7.5) 3 times. Finally, after 6 h of incubation, a lysosome tracker (LysoView 405) was added for live-cell imaging. Z-stack confocal imaging was performed with a Zeiss 780 laser-scanning microscope (Zeiss, Jena, Germany). Nanodiamonds were detected at 561/659 nm, pHrodo Green Dextran at 509/533 nm, and LysoView 405 at 400/464 nm.

## 3. Results and Discussion

### 3.1. FNDs Characterization

FNDs and dextran-coated FNDs were prepared in DEMI water. The results of size and zeta potential measurements are shown in Figure 1a,b. After dextran coating, FNDs did not change in size or zeta potential.

The inside of endosomes and lysosomes is a rather acidic environment where the pH ranges from 6 to 4.5. Thus, we also investigated pH influences on the diamond zeta potential. The results are shown in Figure 1c. With dextran coating, the FND zeta potential slightly increased from −20 mV to −10 mV compared to bare FND.

### 3.2. SEM Imaging

The coating of pHrodo Green Dextran on the diamond surface was investigated by SEM. SEM images in Figure 2 show a thin film on the diamond surface in coated particles but not in bare FNDs. pHrodo Green Dextran T also causes particles to have a more spherical shape.

### 3.3. Cell Viability

Cell viability was tested by the MTT assay. To this end, cells were incubated with different concentrations of bare FNDs (5 µg/mL, 10 µg/mL), different concentrations of FNDs (5 µg/mL, 10 µg/mL) mixed with dextran, 5% DMSO (as a positive control) for 24 h. No significant differences were found between the control and the cells exposed with FNDs. As can be seen from Figure 3, neither dextran nor nanodiamonds used in our experiment were toxic to HeLa cells.

### 3.4. Cellular Uptake of FNDs

One aim of the project was to determine the fate of diamond particles inside HeLa cells. However, bare FNDs are barely taken up by HeLa cells. As a result, it is difficult to compare diamond signal with stained endocytosis vehicles. Surprisingly, with pHrodo Green Dextran coating on the diamond surface, there are significantly more diamond particles inside HeLa cells (besides allowing them to follow the endocytic pathway). Even for a short incubation time of 30 min we already observed uptake (see Figure 4a for confocal images, Figure 4b for statistical analysis, *p* < 0.0001). With pHrodo Green Dextran coating, diamond particles aggregate slightly in HeLa cells. These aggregates are around 1 µm in diameter (see Appendix A).

Similar increase in uptake has already been reported for other particles before. Coating nanoparticles with mannosylated dextran enhanced antigen delivery to dendritic cells (DCs) [32]. Another study also showed that positively charged dextran–spermine nanoparticles can interact with the anionic substances on the cell surface and thus can be used for gene delivery [33]. Additionally, Yu et al. used cross-linked dextran–lipoic acid derivative nanoparticles, which were able to cross cell membranes and to translocate DOX into the nuclei of cancer cells (HeLa and RAW264.7) [34]. Dextran is also a classic endocytic tracer; J Fermie et al. showed fluorescent bovine serum albumin distributed in the endo-lysosomal system in cells by using dextran488 labelling [35].

In order to find out if dextran plays a role in improving the uptake of nanodiamonds, control groups (bare FNDs, cells first blocked with dextran then incubated with bare FNDs, dextran mixed with FNDs) were used to test our hypothesis. For the control groups, dextran was used in the same concentration as in the pHrodo Green Dextran experiments. The final FND concentration was the same as well. Figure 4b shows no difference between bare FND and dextran block cell first in terms of diamond number per cell after 30 min incubation. Dextran plus FNDs slightly increased the number of diamonds in HeLa cells, *p* < 0.05. pHrodo Green Dextran-adsorbed FNDs strongly increased the number of diamonds in HeLa cells, *p* < 0.0001. At longer incubation times (4 h) dextran-blocked cells contained significantly more diamonds *p* < 0.001. Both dextran plus FNDs or pHrodo Green Dextran plus FNDs greatly improved the cellular uptake of diamond particles.

### 3.5. Subcellular Location of FNDs

Endosomes and lysosomes (endolysosomes) are vital organelles, which are responsible for particle uptake. Their acidic (pH is around 4.5 to 5.0 in the lysosomes) interior helps the cells to process and digest. They are also needed for membrane resealing [36], and apoptosis through mitochondrial destabilization [37,38]. Studies showed that endolysosomes’ structural and functional changes can play a role in neurodegenerative diseases and cancer [39,40]. Thus, endolysosomes are always a target for drug delivery and release as well as sensing. In the recent decades, nanoscale carriers show great potential in endolysosomes targeting for treating cancer [34,41] Alzheimer’s disease (ADs) [42], Parkinson’s disease [43], and lysosomal storage diseases [44].

The endocytosis pathway plays a crucial role in cellular uptake of nutrients, drug delivery, cell signalling and cell membrane degradation. Endocytosis is a complicated process by endosomes (early and late endosomes) and lysosomes. Endosomes and lysosomes are acidic due to the electrogenic vacuolar H+ ATPase (also known as V-ATPase), which pumps protons into the endosomal/lysosomal lumen [45]. This acidification of endosomes provides a way to monitor endocytosis or phagocytosis with pH-sensitive probes.

pHrodo Green Dextran is a pH-sensitive fluorescent dye conjugate, which is designed for the detection of phagocytosis and endocytosis. It is almost colourless outside the cell at neutral pH and emits green fluorescence in an acidic environment which increasing when the pH lowers. Interestingly, we found that the pHrodo Green dye increases nanodiamond uptake in HeLa cells even in a very short incubation period (control group didn’t find any FNDs signal Appendix A). The dye remained pH-sensitive after being adsorbed with FNDs (Appendix A). Thus, pHrodo Green Dextran is a useful probe to investigate diamond transportation via endosome vehicles.

Confocal images shown in Figure 5a,b show the signal of pHrodo Green Dextran presented used a Fire lookup table made by FIJI, indicating the change of fluorescent intensity. The fluorescent intensity gradually increased from 1 h incubation to 6 h of incubation, and further greatly increased from 14 h to 72 h. We assume that diamond particles were first in early endosomes which then developed or fused into late endosomes from 1 h incubation until 6 h incubation. From 3 h of incubation, there are late endosomes emerging, which remain present until 14 h. Starting from 14 h incubation, diamond particles are accumulated in lysosomes. This can be seen until 48 h of incubation. This observation can be strongly confirmed by Figure 5c where the green signal of pHrodo Green Dextran is getting brighter as the incubation timing increases. During this time, the diamond signal completely overlapped with pHrodo Green Dextran, indicating that diamonds colocalized very well with endosomes. At 24 or 48 h of incubation, diamonds signals were overlapped with lysosomes signal (in blue).

Additionally, more diamond particles can be found in HeLa cells over time until 6 h of incubation. From 14 h, diamond particles can be found also outside cells. At 48 h incubation, there are significantly fewer particles in cells compared to 24 h of incubation. At the same time, more particles were found outside cells. In addition, at 72 h, there were only very few diamonds remaining inside HeLa cells. We assume that cells were expelling diamonds from 14 h by exocytosis.

## 4. Conclusions

We address the problem of cellular uptake of diamond particles and developed a method to tackle this problem. We demonstrate that dextran in combination with the positively charged dye can enhance uptake dramatically in HeLa cells, which paves a way for future research. Furthermore, pHrodo Green Dextran conjugates allow the monitoring of the local pH around diamond particles. The traffic of pHrodo Green Dextran diamond particles in Hela cells occurs via endosomes and lysosomes. Eventually, the diamond particles are secreted by cells in the end.

Our findings paved a way for effective delivery of drug-loaded diamonds subcellular targeting in the biomedicine field.

## 5. Statistical Analysis

Prism GraphPad 8 was used to analyse data unless indicate, one way ANOVA was used to compare data among groups. A T-test was used to compare between two groups. The data were presented as median with interquartile range and * *p* < 0.05, ** *p* < 0.001, *** *p* < 0.0001 were defined as significant differences.

## Figures and Tables

**Figure 1 nanomaterials-11-01837-f001:**
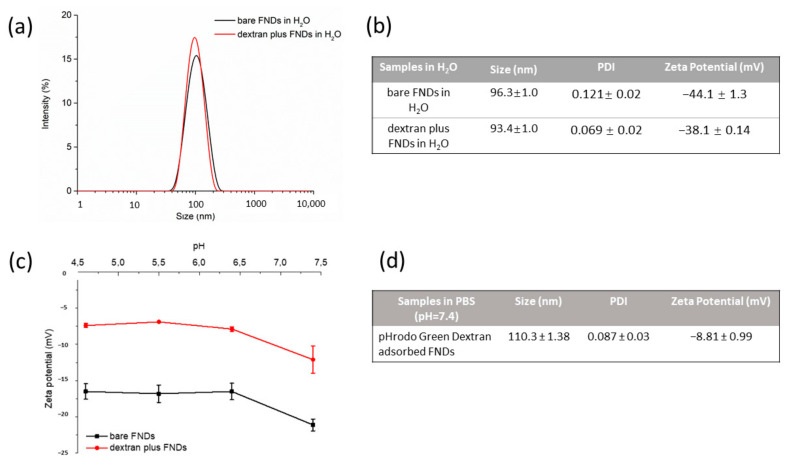
Characterization of 70 nm diamond particles. (**a**) Size distribution of bare FNDs and dextran-coated FNDs in water. (**b**) Size and zeta potential of bare FNDs and dextran-coated FNDs in water. (**c**) Zeta potential of bare FNDs and dextran coated FNDs in PBS at different pH. (**d**) Size and zeta potential of pHrodo Green dextran adsorbed FNDs in PBS at pH 7.4.

**Figure 2 nanomaterials-11-01837-f002:**
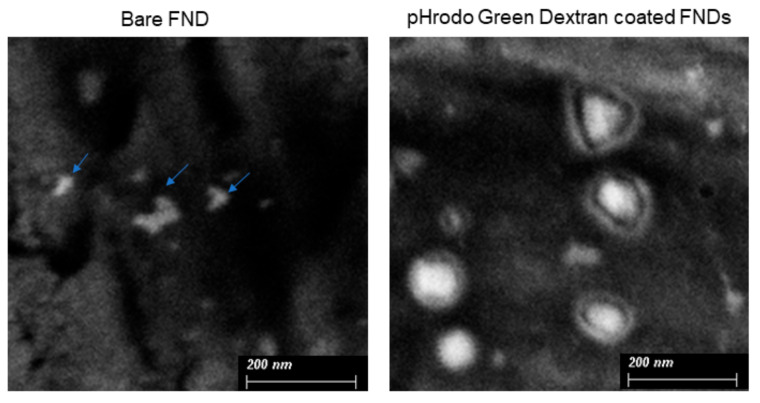
SEM images of bare FNDs and pHrodo Green Dextran adsorbed FNDs from a representative area.

**Figure 3 nanomaterials-11-01837-f003:**
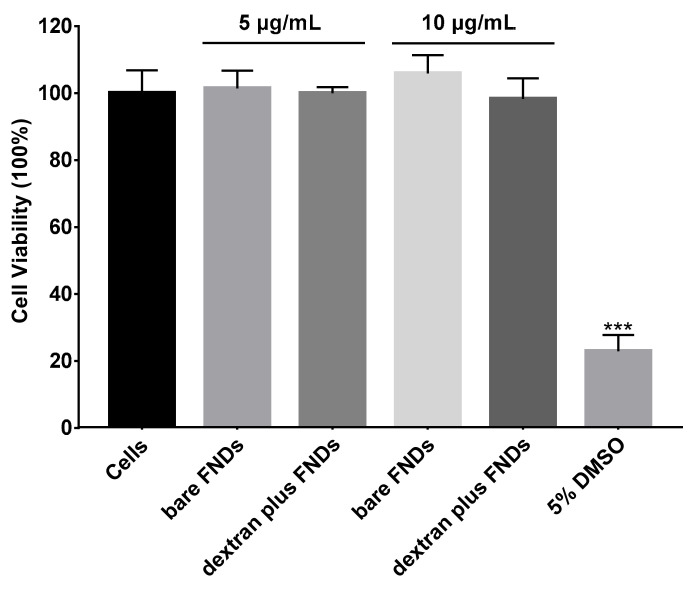
Cell viability test by MTT assay. The FND concentration is 5 or 10 µg/mL. 100% represents a control with no exposure to FNDs. The experiment was repeated three times, error bars represent the standard deviations. (*** *p* < 0.0001).

**Figure 4 nanomaterials-11-01837-f004:**
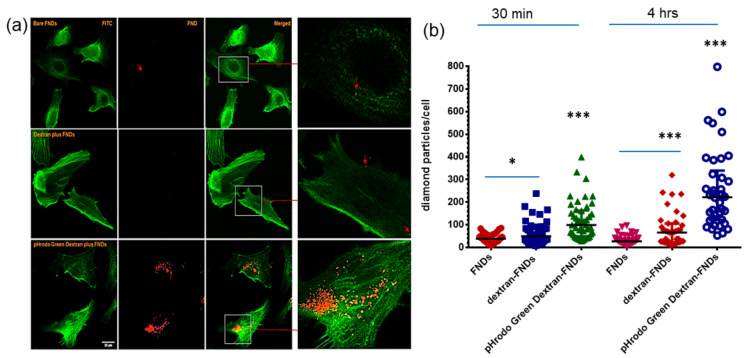
Cellular uptake of diamond particles. (**a**) Confocal images of HeLa cells uptake FNDs, cells incubated with diamond particles for 4 h. green: pHrodo Green Dextran, DIC: Differential interference contrast, red: FNDs, the arrow indicates a diamond position diamond signal, scale bar: 20 µm. Arrows and zoomed in images used to indicate diamond location. (**b**) Analysis of diamond numbers in cells incubated with diamond particles for 30 min and 4 h. Three independent experiments were performed, 100 cells were counted for each group: bare FNDs, dextran plus FNDs, dextran-blocked cells first, pHrodo Green Dextran plus FNDs. * *p* < 0.05, *** *p* < 0.0001 as significant differences.

**Figure 5 nanomaterials-11-01837-f005:**
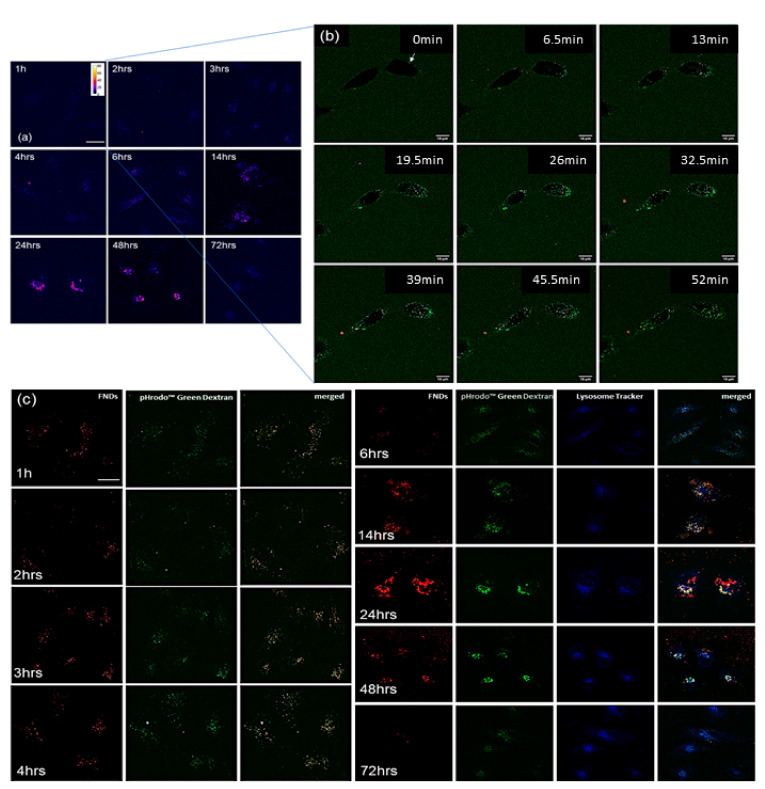
Intracellular location of FNDs in HeLa cells. (**a**) Confocal images of pHrodo Green Dextran adsorbed FNDs incubated with cells for different times. Signals of pHrodo Green Dextran and FNDs are shown in blue using a Fire lookup table in FIJI to investigate the strength of fluorescent intensity changes based on pH changes inside cells. The scale bar is 20 µm. (**b**) Confocal images after 1 h after entry of pHrodo Green Dextran adsorbed FNDs to cells. Bright dots in cells are pHrodo Green Dextran adsorbed FNDs, scale bar 10 µm. Arrows are used to highlight diamond particles. (**c**) Subcellular location of pHrodo Green Dextran adsorbed FNDs. Green: pHrodo Green Dextran, red: FNDs, blue: LysoTracker (Lysoview 405, lysosome tracker), scale bar 20 µm.

## Data Availability

The data presented in this study are available in [insert article or Appendix A here].

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
