# Peer review of "pH Sensitive Dextran Coated Fluorescent Nanodiamonds as a Biomarker for HeLa Cells Endocytic Pathway and Increased Cellular Uptake"

_nanomaterials, 2021, doi:10.3390/nano11071837_

Round 1

Reviewer 1 Report

The authors described the coating of nanodiamonds with a pH-sensitive dye and dextran in order to to increase their uptake by non-phagocytic cells and to follow their endocytosis process.

Questions:

1.Have the authors any other evidence of dextran absorption onto the nanodiamons besides the slight decrease in the zeta potential? Have the authors optimized the amount of dextran adsorbed onto the Nanodiamons=

2.Have the authors evidence of pH dye adsorption onto the nanodiamons? The measure of zeta potential could be helpful for this characterization

3. Is possible the absorption of the dye directly onto the nanodiamons? what is the role of dextran in the system? The uptake of pHrodo Gren-FNDs should be also analysed if this system is feasible.

-Please indicate the meaning of NV centres (in the introduction)

-Replace heading of results by results and discussion

-Replace heading of discussion by conclusion

Author Response

The authors described the coating of nanodiamonds with a pH-sensitive dye and dextran in order to increase their uptake by non-phagocytic cells and to follow their endocytosis process.

Questions:

1.Have the authors any other evidence of dextran absorption onto the nanodiamons besides the slight decrease in the zeta potential? Have the authors optimized the amount of dextran adsorbed onto the Nanodiamons=

Besides zeta potential, we performed DLS to measure the size differences before and after dextran absorption with diamonds, we didn’t see any differences of diamonds’ size. We also performed SEM (Figure 2) to evidence that pHrodo green dextran absorbed on diamond surface.

We didn’t optimize the amount of dextran adsorbed onto nanodiamonds, the concentration of dextran we used was recommended by the manufacturer.

2.Have the authors evidence of pH dye adsorption onto the nanodiamons? The measure of zeta potential could be helpful for this characterization

We added DLS and zeta potential data of pHrodo Green dextran adsorbed nanodiamonds in Figure 1 (d).

  1. Is possible the absorption of the dye directly onto the nanodiamons? what is the role of dextran in the system? The uptake of pHrodo Gren-FNDs should be also analysed if this system is feasible.

It is possible that the dye directly adsorbed onto nanodiamonds, the dye has a positive charge, and diamond has a negative charge on the surface, so they bind together by electrostatic interactions.

 pHrodo Green Dextran is a pH sensitive dye conjugated with dextran, dextran is a classic tracer for mapping the endocytosis pathway, together this pHrodo Green Dextran conjugate enhance diamond particles uptake by HeLa cell, and also enable us to track diamond movement via endocytosis pathway.

We investigated pHrodo Green-FNDs uptake by HeLa cells as shown in Figure 4, both confocal images (a) and particles analysis (b) confirmed that pHrodo Green-FNDs enhance diamond uptake.

-Please indicate the meaning of NV centres (in the introduction)

We indicate NV centres as nitrogen-vacancy centers in the introduction.

-Replace heading of results by results and discussion

done

-Replace heading of discussion by conclusion

done

Reviewer 2 Report

Authors found that the pHrodo Green dye increases nanodiamond uptake in HeLa cells.  This result is very important for nanodiamond particles application in nanomedicine. But there are several important points and questions concerning the sumbitted manuscript/results. Using  fluorescence nanodiamonds  as a marker in cells, it is necessary to discuss the immune response of the cell to the presence of such stable inorganic nanoparticle: T Svadlakova, et al, Nanomaterials 10 (3), 418.  Beside that authors should propose possible explanation of the important and application relevant finding that the pHrodo Green dye increases nanodiamond uptake in HeLa cells. 

Author Response

Authors found that the pHrodo Green dye increases nanodiamond uptake in HeLa cells.  This result is very important for nanodiamond particles application in nanomedicine. But there are several important points and questions concerning the sumbitted manuscript/results. Using  fluorescence nanodiamonds  as a marker in cells, it is necessary to discuss the immune response of the cell to the presence of such stable inorganic nanoparticle: T Svadlakova, et al, Nanomaterials 10 (3), 418.  Beside that authors should propose possible explanation of the important and application relevant finding that the pHrodo Green dye increases nanodiamond uptake in HeLa cells.

We added more explanation of possible applications and working mechanism related with pHrodo Green dye increased nanodiamond uptake in HeLa cells.

Round 2

Reviewer 1 Report

None, the authors have kindly replied the questions